# Difunctionalization of Dienes, Enynes and Related Compounds via Sequential Radical Addition and Cyclization Reactions

**DOI:** 10.3390/molecules28031145

**Published:** 2023-01-23

**Authors:** Sanjun Zhi, Hongjun Yao, Wei Zhang

**Affiliations:** 1Jiangsu Key Laboratory for the Chemistry of Low-Dimensional Materials, Huaiyin Normal University, Huai’an 223300, China; 2College of Biological Science and Technology, Beijing Forestry University, 35 Qinghua East Road, Beijing 100083, China; 3Department of Chemistry, University of Massachusetts Boston, 100 Morrissey Boulevard, Boston, MA 02125, USA

**Keywords:** radical, difunctionalization, addition, cyclization, diene, enyne

## Abstract

Radical reactions are powerful in creating carbon–carbon and carbon–heteroatom bonds. Designing one-pot radical reactions with cascade transformations to assemble the cyclic skeletons with two new functional groups is both synthetically and operationally efficient. Summarized in this paper is the recent development of reactions involving radical addition and cyclization of dienes, diynes, enynes, as well as arene-bridged and arene-terminated compounds for the preparation of difunctionalization cyclic compounds. Reactions carried out with radical initiators, transition metal-catalysis, photoredox, and electrochemical conditions are included.

## 1. Introduction

Synthetic radicals are a topic of current interest due to their feasible radical transformations, such as addition, cyclization, coupling, atom/group transfer, rearrangement and fragmentation, which are powerful in the construction of carbon–carbon bonds, carbon–heteroatom bonds and the formation of diverse ring skeletons [1,2]. The recent developments on photoredox catalysis [3] and electrochemical reactions [4] have sped up the research in this field. Among the board scope of free radical reactions, the radical difunctionalization of alkenes and alkynes has attracted special attention since the substrates are readily available, the reaction process is operationally simple, and two functional groups are introduced to the products in regio- and diastereoselective fashions.

There is a large number of reviews on the radical difunctionalizations [5,6,7,8,9,10,11,12,13,14,15,16]. In a recent paper from our group, we summarized radical addition followed by nucleophilic addition for 1,2- and remote difunctionalizations to introduce X and Y groups to the products (Figure 1, I) [17]. Presented in this paper is another kind of radical difunctionalization that is initiated with the addition of radical X^.^ followed by radical cyclization and then a second functionalization with Y through coupling or addition to obtain the product (Figure 1, II). 

More information on the radical addition and cyclization-based difunctionalization reactions is shown in Figure 2. There are three different kinds of substrates: (I) dienes, diynes, and enynes; (II) arene-tethered dienes or enynes; and (III) arene-terminated alkenes and alkynes. The cyclized radical intermediates could have four ways for the second functionalization with Y: (I) coupling with radical Y; (II) metal-catalyzed reaction with Met-Y; (III) oxidation to a cation and then undergoing nucleophilic reaction with Y^−^; and IV) reduction to an anion and then undergoing electrophilic reaction with Y^+^. The difunctionalization reactions could be carried out as a one-pot reaction with the following cascade reaction sequence: (1) addition of the initial radical X^.^ to introduce the first functional group; (2) radical cyclization to form the ring; and (3) second functionalization with Y to obtain the product. Compared to the two kinds of reactions shown in Figure 1, the first one is relatively simple and has been well established. The second one can generate structurally more attractive fused-, bridged-, or spiro-ring systems, but they are more synthetically challenging and under active development. The reactions presented in this paper are organized based on three kinds of starting materials shown in Figure 2. Reaction-related substrates are also discussed in the last section of the paper.

The radical reactions presented in this paper could be conducted using one of the following methods: (1) using radical initiators such as azodiisobutyronitrile (AIBN), *t*-butyl nitrite (*t*-BuONO), aryldiazonium tetrafluoroborates; peroxides such as dicumyl peroxide (DCP), di-*t*-butyl peroxide (DTBP), and *t*-butyl hydroperoxide (TBHP) and *t*-butyl peroxybenzoate (TBPB); (2) using single electron transfer (SET) agents such as hypervalent iodine reagents (HIRs), hypervalent bromine reagents (HBrRs), ceric ammonium nitrate (CAN), Mn(OAc)_2_, and Na_2_S_2_O_5_; (3) under photoredox catalysis such as Ru(bpy)_3_Cl_2_ and Ir(ppy)_3_), [Ir(dtbbpy)(ppy)_2_]PF_6_, *N*-methyl-9-mesityl acridinium (Mes-Acr^+^), *fac*-Ir(ppy)_3_, Na_2_-Eosin Y; and (3) through electrochemical reactions. 

A wide range of functional groups could be incorporated to the products through the difunctionalization reactions, which include halogens (Cl, Br and I), aryl (Ar), alkyl (R), cyano (CN), trifluoromethyl (CF_3_) or perfluoroalkyl (R_F_), 2-ethoxy-1,1-difluoro-2-oxoethyl (CF_2_CO_2_Et) or 2-ethoxy-1-fluoro-2-oxoethyl (CHFCO_2_Et), 2-cyanopropan-2-yl (C(CH_3_)_2_CN), carbamoyl (CONH_2_), aryl carbonyl (ArCO), alkyl carbonyl (RCO), hydroxy (OH), carbamoyl oxy (O_2_CNR_2_), azido (N_3_), amino (NR_2_), aryldiazenyl (Ar-N=N), nitro (NO_2_), nitroso (NO), sulfonyl (Ts), trifluoromethylthio (CF_3_S), methylthio (CH_3_S), arylthio (ArS), phosphorus (PO(OR)_2_), alkyl silyl (R_3_Si), aryl silyl (Ar_3_Si), phenylselanyl (PhSe) and heteroatom-containing groups.

## 2. Reaction of Dienes and Enynes

Presented in this section are the radical addition and cyclization-initiated difunctionalization reactions of 1,n-dienes and -enynes with a reaction sequence shown in Figure 3. The common substrates include dienes (**I-A**), enynes (**I-B**, **I-C**, **I-H**), dienyl amides (**I-F**), enynyl amides (**I-D**, **I-E**, **I-G**) with the Z as a carbon or heteroatom (Figure 4). Since there are two unsaturated carbon–carbon bonds in the substrates which are available for the radical addition, the regioselectivity for the initial radical addition is critical. As indicated in Figure 4, the steric hindrance (**I-A** to **I-D**) and conjugation effect of the groups, such as C=O and Ar (**I-E** to **I-H**), are the major factors to direct the position for the initial radical addition.

In 2005, Ogawa and coworker reported a near-UV light-mediated radical reaction of dienes, diynes, and enynes for the synthesis of iodoperfluoroalkylated cyclic products. The reactions of dienes, diynes, or enynes and perfluoroalkyl iodides in PhCF_3_ under the irradiation of xenon lamp afforded products **1** as a mixture of *cis/trans* isomers in moderate-to-good yields (Figure 5) [18]. A proposed mechanism indicated that the *n*-C_4_F_9_ radical generated from *n*-C_4_F_9_I under the light adds to diene. The intermediate **M-1** undergoes 5-*exo* cyclization to give alkenyl **M-2,** which then reacts with *n*-C_4_F_9_I through the iodine atom transfer to give product **1a**.

A sun lamp-mediated radical reaction for making azidosulfonylated cyclic products was reported by the Renaud group in 2008. Dienes, diynes, or enynes in dry benzene reacted with benzenesulfonyl azide with radical initiator di-*t*-butyldiazene to give azidosulfone products **2** in moderate-to-excellent yields (Figure 6) [19]. This method is good for the formation of tertiary and secondary azides **2a–d**, but not for primary azide **2e**. The reaction process involves the addition of PhSO_2_ radical to the less hindered alkene to form intermediate radical **M-3**, 5-*exo* cyclization for radical **M-4**, and N_3_ radical transfer from PhSO_2_N_3_ to give product **2a**.

1,6-Enynes are the most popular substrates for radical reactions to make difunctionalized five-membered rings. A method for making iodotriflouromethylated *N*-heterocycles was reported by the Liu group in 2014. The reaction of 1,6-enynes, NaSO_2_CF_3_ and I_2_O_5_ in CH_2_Cl_2_/H_2_O afforded pyrrolidines products **3** in moderate-to-high yields (Figure 7) [20]. The CF_3_ radical generated from NaSO_2_CF_3_ through SET of I_2_O_5_ adds to the alkenyl group of 1,6-enynes followed by cyclization and the capture of iodine to give products **3**. The CF_3_ radical could be trapped by 2-methyl-2-nitrosopropane (MNP) to form **M-5** for ESR detection.

A method for cyclative trifluoromethylation of 1,6-enynes was reported by the Liang group in 2014. The reaction of 1,6-enynes, Togni’s reagent, and TMSCN (or TMSN_3_) in CH_3_CN under the catalysis of Cu^II^ gave CF_3_-containing heterocycles **4** and **5** (Figure 8) [21]. The CF_3_ radical produced from the Togni’s reagent under the catalysis of Cu^II^ adds to the C=C double bond of 1,6-enyne to form the radical intermediate **M-6,** which is converted to cyclized metal complex **M-7** through path a or path b. At the last step, the reaction of **M-7** with TMSCN or TMSN_3_ gives corresponding cyanotrifluoromethylated or azidotrifluoromethylated five-membered ring products **4a** or **5a**.

A Togni’s reagent-based synthesis of CF_3_-substituted spiro 2*H*-azirines was reported by the Liang group in 2015. The reaction of 1,6-enynes with Togni’s reagent and TMSN_3_ in the presence of Cu^0^ powder as a catalyst afforded diastereomeric products **6** in good-to-excellent yields (Figure 9) [22]. A proposed mechanism suggests that the CF_3_ radical generated from Togni’s reagent through SET of Cu^0^ is added to the C=C bond of 1,6-enyne to produce the radical intermediate **M-8.** Sequential 5-*exo* cyclization and trapping of the radical **M-9** with Cu^II^ and TMSN_3_ give Cu^II^ azide complex **M-10**. Complex **M-10** may also be obtained from the formation of complex **M-11** and subsequent cyclization. Reductive elimination of **M-10** followed by the elimination of N_2_ from azide **M-12** gives alkenyl nitrene **M-13**. The cyclization of **M-14,** a resonance structure of alkenyl nitrene **M-13**, gives the spiroketal products **6** as a pair of diastereomers.

Liang’s lab introduced a method for Pd-catalyzed radical cyclative iododifluoromethylation of 1,6-enynes in 2015. The reaction of 1,6-enynes and ethyl difluoroiodoacetate in dioxane under the catalysis of Pd(PPh_3_)_2_Cl_2_ and bis-[2-(diphenyl-phosphino)phenyl]ether (DPE-Phos) gave iododifluoromethylated heterocycles **7** in good-to-excellent yields (Figure 10) [23]. The CF_2_CO_2_Et radical is generated from ICF_2_CO_2_Et through the reduction of Pd^0^L_n_. Radical addition to the C=C double bond of 1,6-enynes followed by the cyclization to Pd^I^L_n_I-activated alkyne group and reductive elimination of the Pd^0^L_n_ gives iododifluoromethylated products **7**.

A sulfonyl radical-initiated iodosulfonylation reaction of 1,6-enynes was reported by the Liang group in 2016. The reaction of 1,6-enynes and sulfonyl hydrazide in the presence of I_2_/TBHP gave five-membered heterocycles **8** in good-to-excellent yields (Figure 11) [24]. 

A proposed mechanism indicated that the sulfonyl radical generated from the reaction of sulfonyl hydrazide and TBHP adds to the C=C double bond of 1,6-enyne, followed by the radical cyclization and coupling with iodine radical, to give product **8a**.

In 2018, the Liang group introduced radical cyclization of 1,6-enynes for the synthesis of substituted pyrrolidine derivatives. The reaction of 1,6-enynes, ICF_2_CO_2_Et in the presence of *N*-methylpiperidine or borophenylic acids/K_2_CO_3_ afforded substituted pyrroles **9** or **10** in moderate-to-good yields (Figure 12) [25]. The initial CF_2_CO_2_Et radical generated from the reaction of ICF_2_CO_2_Et adds to the C=C double bond of the 1,6-enyne followed by 5-*exo* cyclization to give radical intermediate **M-15**. Radical **M-15** abstracts iodo atom from iododifluoromethylation to give product **9a**; otherwise, coupling of **M-15** with borophenylic acid gives product **10a**.

A visible light-mediated radical sulfonylative and azidosulfonylative cyclization of 1,6-enynes for the synthesis of highly functionalized heterocycles was introduced by the Lam group in 2017. The reaction of 1,6-enynes and sulfonyl azides in THF in the presence of a photoactive iridium complex afforded difunctionalized heterocycles **11** or **12** in moderate-to-excellent yields (Figure 13) [26]. The use of THF as the solvent was critical for the success of the reactions. The reaction mechanism suggests that the sulfonyl radical generated from TsN_3_ under the visible light catalysis of [Ir(dtbbpy)(ppy)_2_]PF_6_ adds to the triple bond of 1,6-enyne, followed by cyclization of the vinyl radical, giving six-membered tertiary radical **M-16**. Product **11a** is then obtained via azidation of **M-16** with the arylsulfonyl azide and the sulfonyl radical is regenerated. When R^1^ is H, addition of the sulfonyl radical happens at the terminal carbon of the triple, followed by cyclization of the vinyl radical to give five-membered ring product **12a.**

The Wu group, in 2017, introduced a reaction of 1,6-enynes with DABCO·(SO_2_)_2_ and two equivalents of ArN_2_BF_4_ in DCE to give diazosulfonated six-membered heterocycles **13** in moderate-to-good yields (Figure 14) [27]. Five-membered heterocycles **14a** could be obtained using unsubstituted terminal alkynes as the substrates. The reaction mechanism suggests that the initially sulfonyl radicals, generated from the reaction of ArN_2_BF_4_ with DABCO·(SO_2_)_2_, adds to the C≡C bond of 1,6-enynes to form vinyl radical **M-18,** followed by 6-*exo* cyclization and trapping with aryldiazonium cation to give intermediates **M-19**. The last step SET of arylsulfonyl radical or DABCO·(SO_2_)_2_ to radical **M-19** gave products **13**.

The Xu group, in 2018, introduced a visible light-mediated radical atom transfer radical cyclization (ATRC) of 1,6- and 1,7-enynes for the synthesis of sulfonyl and trifluoromethylthio functionalized vinylsulfones. In the ATRC reactions, two functional groups are from the same reagent. The reaction of enynes and PhSO_2_SCF_3_ in the presence of PPh_3_AuNTf_2_ and Ru(bpy)_3_Cl_2_ under the irradiation of blue LED afforded five- or six-membered vinylsulfones **15** in good yields (Figure 15) [28]. A proposed mechanism for the reaction of 1,6-enyne indicated that the sulfonyl radical generated from PhSO_2_SCF_3_ under photocatalysis of PPh_3_AuNTf_2_ and Ru(bpy)_3_Cl_2_ adds to the triple bond to form benzyl radical **M-20,** followed by 6-*exo* cyclization to give tertiary radical **M-21**. It then couples CF_3_S radical to give product **15a**. For the reaction of a 1,6-enyne without substitution on the terminal carbon (R^1^ = H), sulfonyl radical adds to the terminal carbon of alkyne followed by 5-*exo* cyclization, leading to product **15d**. A similar process for the reaction of 1,7-enyne, which has no terminal carbon substitution on alkyne, affords product **15e**.

A visible light-mediated ATRC of 1,6-enyne for the preparation of chloroalkyl-substituted cyclic alkenyl sulfones using sulfonyl chlorides as the key reactants was reported by the Zhu group in 2018. The reactions of 1,6-enynes and sulfonyl chlorides in the presence of [Ir(dtbbpy)(ppy)_2_]PF_6_ under the irradiation of blue LED gave five- or six-membered chloroalkyl-substituted cyclic alkenyl sulfones **16** or **17** (Figure 16) [29]. As the reaction mechanism indicated, the sulfonyl radical generated from TsCl under the photoredox of [Ir(dtbbpy)(ppy)_2_]PF_6_ adds to the C≡C bond of the 1,6-enyne followed by 5-*exo* or 6-*exo* cyclization to form the carbon radicals **M-24** or **M-25**. They are oxidized to carbocations **M-26** and **M-27** and then react with chlorine anion to form products **16** and **17**, respectively. 

In 2018, the Liu group reported the synthesis of bromotrifluoromethylated five- and six-membered heterocycles. The reaction of 1,6- or 1,7-enynes, NaSO_2_CF_3_ and NaBrO_3_ in DCM/H_2_O produced products **18** in good yields (Figure 17) [30]. The CF_3_ radical, generated from the reaction of NaSO_2_CF_3_ and NaBrO_3_, adds to the terminal carbon of alkene followed by 5-*exo* or 6-*exo* cyclization (n = 2) and then Br-atom abstraction to give product **18**.

Lin and coworkers reported an electrochemical reaction for the preparation of chlorotrifluoromethylated pyrrolidines in 2018. The reaction was carried out using HOAc-MeCN as solvent at room temperature under electrochemical conditions. The reaction of 1,6-enynes, CF_3_SO_2_Na and MgCl_2_ in the presence of LiClO_4_ and Mn(OAc)_2_ gave chlorotrifluoromethylated pyrrolidines **19** in excellent yields (Figure 18) [31]. The initial CF_3_ radical generated from the anodically coupled electrolysis adds to the C=C double bond of 1,6-enynes followed by 5-*exo* cyclization to afford the vinyl radical **M-28,** which couples with the Cl radical to give product **19**. 

A visible light-promoted reaction of 1,6-enynes for the synthesis of difunctionalized pyrrolidines was introduced by the Wang group in 2020. The reaction of 1,6-enynes, and chalcogens (such as benzenesulfono–selenoate) in acetone at room temperature under the radiation of blue LED afforded products **20** in moderate-to-good yields (Figure 19) [32]. The reaction mechanism suggests that tosyl and phenylselenyl radicals are generated from Se-phenyl 4-methylbenzenesulfonoselenoate under photo irradiation. The tosyl radical adds to the C=C bond of 1,6-enyne followed by 5-*exo* cyclization and capture of phenylselenyl radical to give product **20a**.

An iodine radical-initiated reaction for the synthesis of difunctionalized *N*-heterocyclic compounds was reported by the Wang group in 2020. The reactions of 1,6- or 1,7-enynes, TBHP and I_2_ in CH_3_CN gave compound **21** in moderate-to-good yields (Figure 20) [33]. The reaction mechanism suggests that iodide radical, generated from the reaction of I_2_ with TBHP, adds to the C≡C triple bond of enyne followed by 6-*exo* cyclization to yield tertiary radical **M-29**. Addition of hydroxyl radical or *t*-butylperox radical to **M-29** could lead to the formation of product **21a.**

In 2021, Zhu and co-workers reported the synthesis of iodo- and nitro-functionalized cyclic compounds such aspyrrolidines, tetrahydrofurans, and cyclopentanes. The reaction of 1,6-enynes, *t*-BuONO, and iodoform in CH_3_CN under heating gave five-membered heterocycles **22** in moderate-to-excellent yields (Figure 21) [34]. The reaction mechanism suggests that nitroso radical formed from the homolysis of *t*-BuONO adds to the C=C bond of the 1,6-enyne followed by 5-*exo* cyclization, oxidation to cation, and then iodination with CHI_3_ to give product **22a**.

In 2021, Zhu and co-workers reported diarylselenylative cyclization reaction of 1,6-enynes for the synthesis of five-membered heterocycles. The reaction of 1,6-enyne and diaryldiselane in toluene under the radiation of light at room temperature afforded products **23** in moderate to good yields (Figure 22) [35]. The reaction mechanism shows that the PhSe radical generated via photo homolytic cleavage of PhSeSePh adds to the triple bond of 1,6-enyne followed by 5-*exo* cyclization to form tertiary carbon radical **M-30**, which then couples with PhSe radical to give product **23a**.

The reaction of 1,6-enynes for the synthesis of dihalogenated pyrrolidines was reported by the Tong group in 2021. The reaction of 1,6-enynes, PhI(OAc)_2_ and lithium halide at room temperature gave product **24** in moderate-to-good yields (Figure 23) [36]. A suggested mechanism for the reaction with LiCl indicated that the Cl radical generated via a single electron oxidation of LiCl with PhI(OAc)_2_ adds to the C=C double bond of 1,6-enyne followed by 5-*exo* cyclization and Cl atom abstraction to give dichloro pyrrolidine **24d**.

In 2021, Li and Tian’s lab reported Fe-catalyzed radical reaction of 1,6-enynes for the synthesis of difunctionalized heterocycles. The reaction of 1,6-enynes, *t*-butyl nitrite (TBN) and KI or NaBr as materials in CH_3_CN under the catalysis of FeSO_4_·7H_2_O gave products **25** in good-to-excellent yields (Figure 24) [37]. As shown in the proposed mechanism, NO_2_ radical produced from TBN adds to the C=C bond of 1,6-enyne followed by 5-*exo* cyclization to give vinyl radical. This radical intermediate is iodinated through two possible pathways to give target product **25a**. 

A Cu-catalyzed radical reaction of 1,6-enynes for the synthesis of cyanoalkylsulfonyl-ated pyrrolidines was introduced by He and coworkers in 2021. The reaction of 1,6-enynes, diselenides, DABCO(SO_2_)_2_ and cyclic ketone oxime esters in DCE with CuOAc as a catalyst afforded functionalized pyrrolidines **26** in moderate-to-good yields (Figure 25) [38]. As indicated in the proposed mechanism, cyanoalkylsulfonyl radical generated from the reaction of cyclic ketone oxime esters and DABCO(SO_2_)_2_ adds to the C=C double bond of 1,6-enyne followed by 5-*exo* cyclization and then couples with PhSe radical to give product **26a**.

In 2019, Zhu and Hou’s group reported a visible light-mediated radical reaction for the synthesis of chlorotrifluoromethylated and chlorotrichloromethylated pyrrolidines, cyclopentanes and related compounds. The reaction of 1,6-enynes and CF_3_SO_2_Cl (or CCl_3_SO_2_Cl) in CH_2_Cl_2_ using Acr^+^-Mes or Ir(dtbbpy(ppy)_2_PF_6_ as a photocatalyst gave products **27** in good-to-excellent yields (Figure 26) [39]. A proposed mechanism indicated that CF_3_ radical generated from CF_3_SO_2_Cl via SET adds to the C=C bond of 1,6-enynes, followed by 5-*exo* cyclization and coupling with Cl radical, to give product **27a**.

In 2022, Li and Yang reported a visible light-promoted reaction of 1,6-enynes for the synthesis of the iodovinyl- and CF_2_-functionalized heterocycles. The reaction of 1,6-enynes, ICF_2_CO_2_Et under the radiation of blue LED afforded products **28** in good-to-excellent yields (Figure 27) [40]. The reaction mechanism suggests that CF_2_CO_2_Et radical derived from ICF_2_CO_2_Et adds to the C=C double bond of 1,6-enyne, followed by 5-*exo* cyclization and capture of iodine atom from ICF_2_CO_2_Et, to give product **28**.

Zhu and co-workers, in 2022, reported a photo synthetic method for making iodo- and sulfonyl-containing cyclic compounds. The reaction of 1,6-enynes, ArSO_2_Na, and iodoform in CH_3_CN under visible light irradiation gave products **29** in good-to-excellent yields (Figure 28) [41]. The reaction mechanism suggests that ArSO_2_ radical derived from ArSO_2_Na adds to the C=C double bond of 1,6-enyne, followed by 5-*exo* cyclization and iodine atom transfer from the complex of ArSO_2_Na and CHI_3,_ to give product **29a**. 

In 2022, a photo reaction of *β*-caryophyllene, a 1,5-diene with one alkene in the ring and another one out of the ring, for the synthesis of iodo- and CF_2_-containing protoilludanes was reported by the Huang group. The reaction of *β*-caryophyllene and ICF_2_COR in the presence of 2-bromophenol and base under the irradiation of blue LED afforded functionalized protoilludanes **30** in excellent yields (Figure 29) [42]. A reaction mechanism suggests that the EDA complex generated from 2-bromophenol and ICF_2_COR leads to the formation of CF_2_COR radical. It then selectively adds to C8 of *β*-caryophyllen, followed by the cyclization and abstraction of iodine atom from ICF_2_COR to give the product **30**.

In 2019, the Liu group reported a met-catalyzed reaction of 1,6-enynes or 1,6-enynyl amides for the synthesis of bromotrihalomethylated pyrrolidines. The reaction of 1,6-enynes, and CCl_3_Br or CBr_4_ in 1,4-dioxane under the catalysis of [Rh(cod)Cl]_2_ and DPE-Phos at 100 °C for 12 h gave products **31** in moderate-to-good yields (Figure 30) [43]. The reaction mechanism suggests that CCl_3_ radical, generated from CCl_3_Br under the catalysis of [Rh(cod)Cl]_2_ and DPE-Phos, adds to the C=C double bond of 1,6-enyne followed by 5-*exo* cyclization to Rh^II^-LBr activated alkyne and then L-Rh^I^ elimination to give product **31a**.

Hou and coworkers, in 2022, reported a Cu-induced radical reaction of 1,6-enynes for the synthesis of functionalized five-membered rings. The reaction of 1,6-enynes, BrCH_2_CN in the presence of CuI, 1,10-phenanthroline and NaHCO_3_ in CH_3_CN afforded products **32** in good yields (Figure 31) [44]. The reaction mechanism suggests that the CH_2_CN radical derived from BrCH_2_CN adds to C=C double bond of 1,6-enyne followed by 5*-exo* cyclization and bromine atom-transfer to give product **32a**.

1,6-Eneynyl amides are another kind of popular substrates for radical reactions in the synthesis of functionalized 2-pyrrolidones [45]. In 2008, Feray and Bertrand reported an R_2_Zn-mediated radical reaction of 1,6-eneynyl amides for the synthesis of functionalized pyrrolidin-2-ones. The reaction of 1,6-eneynyl amides and alkyliodides in the presence dialkylzinc at room temperature gave product **33** in high yields as a mixture of *E/Z* isomers (Figure 32) [46]. The reaction mechanism suggests that the *t*-butyl radical, generated from the reaction of *t*-BuI and R_2_Zn in the presence of oxygen, selectively adds to the triple bond of amide to form a stabilized vinyl radical, which then undergoes 5-*exo* cyclization followed by iodine atom transfer from *t*-BuI to give product **33**.

Xuan and co-workers introduced a reaction of 6-enynyl amides for the synthesis of substituted 2-pyrrolidinones in 2018. The reaction of 6-enynyl amides, NIS (or NBS), and sulfonyl hydrazide in CH_3_CN and in the presence TBHP afforded *γ*-lactams **34** in good to excellent yields (Figure 33) [47]. The reaction mechanism suggests that sulfonyl radical generated from arylsulfonyl hydrazide adds to the C=C double bond of amide followed by 5-*exo* cyclization and then coupling with iodine radical to give product **34a**. 

Wei and co-workers reported a protocol of cyclative chloroazidation of 1,6-enynyl amides for the synthesis of substituted 2-pyrrolidinones in 2018. The reaction of 1,6-enynyl amides, TMSN_3_ and NCS in DCE in the presence of PIDA gave product **35** in moderate yields (Figure 34) [48]. The reaction mechanism suggests that N_3_ and Cl radicals were generated from TMSN_3_ and NCS. The addition of N_3_ radical to the C=C double bond of amide followed by 5-*exo* cyclization and coupling with the Cl radical affords product **35a**.

In 2022, Li and coworkers reported a reaction of 1,6-enynyl amides for the synthesis of *γ*-lactams. The reaction of 1,6-enynyl amides and sulfonyl hydrazides in H_2_O at 70 °C for 20 h in the presence of TBHP gave product **36** in moderate-to-good yields (Figure 35) [49]. The reaction mechanism suggests that PhSO_2_ radical, generated from the reaction of PhSO_2_NHNH_2_ with TBHP and TBAI, adds to the C=C double bond of amide followed by 5-*exo* cyclization and coupling with iodine radical to give product **36a**.

A photoredox ATRC reaction of 1,6-dienyl amides for the synthesis of functionalized pyrrolidin-2-ones was developed by the Miyabe group in 2015. The reaction of 1,6-dienyl amides and iodoalkanes in aqueous media and catalyzed by Ru(bpy)_3_Cl_2_·6H_2_O and (*i*-Pr)_2_NEt gave product **37** in fair-to-good yields (Figure 36) [50]. Other than *i*-C_3_F_7_I, other iodo compounds such ICH_2_CN and ICH_2_CF_3_ are also good radical precursors. The reaction mechanism suggests that the *i*-C_3_F_7_ radical generated from *i*-PrI via the photoredox process adds to the C=C double bond of amide, followed by 5-*exo* cyclization and then iodine atom transfer from *i*-PrI to give product **37a**. 

Li and Wei, in 2021, reported a Cu-catalyzed radical reaction of 1,6-dienyl amides for the synthesis of substituted *γ*-lactams. The reaction of 1,6-dienyl amides and RSO_2_NHNH_2_ in CH_3_CN in the presence of CuI and TBHP gave product **38** in moderate-to-good yields (Figure 37) [51]. The reaction mechanism suggests that the sulfonyl radical, generated from the reaction of RSO_2_NHNH_2_ with TBHP, adds to the C=C double bond of amide followed by 5-*exo* cyclization, oxidation to carbocation, and trapping I^−^ anion of CuI to provide iodosulfonylation of product **38a**. 

A photoredox reaction of carbonyl-containing 1,6-enynes for the synthesis of cyclopentanone derivatives was reported by Zhou, Yu and their coworkers in 2020. The reaction of *gem*-dialkylthio enynes, cyclobutanone oxime esters, and boronic acids in the presence of Cu(CH_3_CN)_4_BF_4_, dtbbpy and K_3_PO_4_ in CH_3_CN under irradiation of blue LED gave functionalized aryl thienyl sulfide **39** in moderate-to-good yields and with good chemo- and diastereoselectivities (Figure 38) [52]. The reaction mechanism suggests that *γ*-cyanoalkyl radical, generated from homolytic *α*,*β*-C−C cleavage of *N*-centered iminyl, which is derived from cyclobutanone oxime esters, adds to the C=C bond of *gem*-dialkylthio 1,3-enyne followed by 5-*exo* cyclization, radical rearrangement and fragment of ethylene to give sulfur-centered radical **M-31**. Radical **M-31** reacts with the LCu^II^Ph complex followed by reductive elimination to give product **39a**.

A reaction of 1,6-enynyl with two carbonyl groups for the synthesis of functionalized succinimides was introduced by the Rong group in 2020. The reaction of 1,6-enynyl amides, NBS or NCS, TMSN_3,_ and PIDA in DCM at room temperature for 3–5 min afforded products **40** as *E/Z* isomers in excellent yields (Figure 39) [53]. The reaction mechanism suggests that the azide radical, resulting from the reaction of PIDA and TMSN_3_, adds to alkene moiety of 1,6-enyne, followed by 5-*exo* cyclization and coupling with the bromine radical from NBS, to give product **40a**.

## 3. Reaction of Arene-Tethered Dienes and Enynes 

Presented in this section are the radical addition and cyclization-initiated difunctionalization reactions of arene-bridged 1,n-dienes -diynes, and -enynes with a reaction sequence shown in Figure 40. It is noteworthy that most substrates found in the literature are enynes but not dienes (like **II-J**) or diynes (like **II-I**) (Figure 41). The enynyl substrates include the most popular 1,7-enynyl amides **II-A** and other ones containing the carbonyl group (**II-B** to **II-E**). Other substrates may contain heteroatom or conjugate groups (such as CN and Ar) at the terminal carbon of the unsaturated bonds (**II-F** to **II-H**). Between the two unsaturated carbon–carbon bonds in the substrates, the regioselectivity for the initial radical addition is directed by the steric and the conjugation effects of the substituents. The R^1^ group on the terminal carbon of alkyne is commonly employed to block the initial radical addition to the alkyne. Substrate **II-J** is an exception in which the initial radical addition does not go to the conjugated alkene.

Benzene-tethered 1,7-enynyl amides are popular substrates for radical difunctionalization reactions. In 2014, the Li group introduced a reaction of such substrates for the synthesis of dinitropyrrolo[4,3,2-*de*]-quinolinones. The reaction of 1,7-enynyl amides and *t*-BuONO in DMSO afforded product **41** in good-to-excellent yields (Figure 42) [54]. It was found that the amount of H_2_O had a significant influence on the reaction. The reaction mechanism suggests that NO_2_ radical generated in situ from *t*-BuONO adds to the C=C double bond of amide followed by 6-*exo* cyclization to form intermediate **M-32**. The reaction of **M-32** with NO or NO_2_ radical followed by electrophilic addition of NO or NO_2_ radical to the phenyl ring gave cationic intermediates **M-33** and **M-34**. Cationic radical intermediates **M-35** and **M-36** were produced through the treatment of the cationic intermediates **M-33** and **M-34** with NO or NO_2_ radical and then lead to the formation of product **41a** after the redox reaction.

The Wu group, in 2016, introduced a photoredox reaction of benzene-tethered 1,7-enynyl amides for the synthesis of trifluoroethyl-substituted 3,4-dihydroquinolin-2(*1H*)-ones. The reaction of 1,7-enynyl amides and Togni’s reagent in the presence of NaI and PhCO_2_H under UV irradiation gave **42** in moderate-to-good yields (Figure 43) [55]. The proposed mechanism indicated that trifluoromethyl radical derived from the Togni’s reagent adds to the C=C double bond of amide, followed by 6-*exo* cyclization and oxidation to cation for the reaction with iodide anion, to give product **42**.

In 2016, the Jiang group reported a reaction of benzene-bridged 1,7-enynyl amides for the synthesis of substituted 3,4-dihydroquinolin-2(1*H*)-ones. The reaction of 1,7-enynyl amides, TMSN_3_ and NIS (or NBS and NCS) in the presence of PhI(OAc)_2_ in CH_2_Cl_2_ gave products **43** in good-to-excellent yields (Figure 44) [56]. A reaction mechanism suggests that N_3_ radical generated from the reaction of PhI(OAc)_2_ and TMSN_3_ adds to the C=C double bond of amide followed by 6-*exo* cyclization and coupling with iodine radical from NIS to give product **43**.

A transition metal-mediated radical reaction of benzene-bridged 1,7-enynyl amides for the synthesis of substituted pyrrolo[3,4-*c*]quinolinones was reported by the Wan group in 2016. The *trans*-fused products were obtained when using Mn^III^ as a catalyst, whereas *cis*-products were obtained using Cu^II^ as a catalyst. The reactions of amides and TMSN_3_ in the presence of Mn(OAc)_3_/NFSI or Cu(ClO_4_)_2_/TBPB in CH_3_CN afforded *trans*- or *cis*-fused products **44,** respectively, in good-to-excellent yields (Figure 45) [57]. A reaction mechanism suggests that N_3_ radical generated from TMSN_3_ adds to the C=C double bond of amides followed by 6-*exo* cyclization, releasing of N_2_, then azido group transfer to afford the desired *trans*- or *cis*-fused product **44**.

The Tu group reported a method for the synthesis of densely functionalized 3,4-dihydro-quinolin-2(1*H*)-ones in 2016. The reaction of benzene-tethered 1,7-enynyl amides, arylsulfonyl hydrazides and NIS (or NBS) in DEC in the presence of TBHP afforded product **45** in good-to-excellent yields (Figure 46) [58]. The reaction mechanism suggests that the sulfonyl radical derived from sulfonyl hydrazides adds to the C=C double bond of amides, followed by 6-*exo* cyclization and coupling with iodine radical from NIS, to give product **45**.

A new method for the synthesis of 3,4-dihydroquinolin-2(1*H*)-ones was reported by the Guo group in 2017. The reaction of benzene-tethered 1,7-enynyl amides, sulfinic acids and diphenyl diselenides in EtOH-H_2_O and in the presence of TBHB to give product **46** in moderate-to-excellent yields (Figure 47) [59]. Carrying out the reaction under micro flow conditions could reduce the reaction time to less than 1 min. The reaction mechanism suggests that the sulfonyl radical, produced from the arylsulfinic acid with the oxidation of TBHP, adds to the C=C double bond of amide followed by 6-*exo* cyclization and coupling with phenylselenyl radical to give product **46a**.

A Cu-catalyzed radical trifluoromethylative spirocyclization reaction of benzene-tethered 1,7-enynyl amides for the synthesis of trifluoromethyl-substituted 1′*H*-spiro-[azirine-2,4′-quinolin]-2′(3′*H*)-ones was introduced by the Han group in 2017. The reaction of amides, Togni’s reagent and TMSN_3_ in DMF and in the presence of Cu^II^ catalyst gave product **47** in good-to-excellent yields (Figure 48) [60]. The reaction mechanism suggests that the CF_3_ radical from Togni’s reagent adds to the C=C double bond of amides; then, it goes through path a or b to give cyclized Cu^III^-azido complex **M-37,** followed by reductive catalyst elimination and denitrogenative cyclization to give product **47**.

The Guo group, in 2019, reported two photoredox methods for the synthesis of trifluoroethyl-substituted 3,4-dihydroquinolin-2(*1H*)-ones. Method 1 is the reaction of 1,7-enynyl amides, CF_3_SO_2_Na, NCS (or NBS) using photocatalyst *N*-methyl-9-mesityl acridinium (Mes-Acr^+^). Method 2 is the reaction of 1,7-enynyl amides and CF_3_SO_2_Cl using photocatalyst *fac*-Ir(ppy)_3_. These two methods gave product **48** in moderate-to-excellent yields (Figure 49) [61]. The proposed reaction mechanism indicated that for method 1, the CF_3_ radical generated from the CF_3_SO_2_Na under the photocatalysis of Mes-Acr^+^ adds to the C=C bond of amide followed by 6-*exo* cyclization and coupling with bromo radical from NBS to give product **48d**. In method 2, the CF_3_ radical generated from the CF_3_SO_2_Cl under the photocatalysis of *fac*-Ir(ppy)_3_ goes through similar addition, cyclization and halogen atom abstraction processes to afford product **48a**. 

A visible light-induced radical reaction for the synthesis of haloperfluorinated *N*-heterocycles was reported by the Tang group in 2019. The reaction of 1,6- or 1,7-enynyl amides, perfluoroalkyl iodides/bromides in 1,4-dioxane and in the presence of *fac*-Ir(ppy)_3_ and K_3_PO_4_ under blue LED irradiation afforded product **49** in good yields and stereoselectivity (Figure 50) [62]. The reaction mechanism suggests that *n*-C_4_F_9_ radical generated under the photocatalysis with of *fac*-Ir(ppy)_3_ adds to the C=C bond of amide, followed by 6-*exo* cyclization and coupling with iodine radical, to selectively give product **49a** as the *Z*-isomer.

The Andrade group reported an ultrafast Fe-promoted reaction for the synthesis of 2-quinolinone-fused *γ*-lactones in 2021. The reaction of benzene-tethered 1,7-enynyl amides and formamide and Fenton’s reagent under microwave irradiation for 10 s gave product **50** in a good overall yield (Figure 51) [63]. The reaction mechanism suggests that the hydroxyl radical generated from Fenton’s reaction adds to the C=C double bond of amide followed by 6-*exo* cyclization, coupling with hydroxyl radical, epoxidation, and lactonization to give product **50a**.

In 2022, Wu, Ying and their coworkers introduced a Pd-catalyzed reaction for the synthesis of perfluoroalkyl and carbonylated 3,4-dihydroquinolin-2(1*H*)-ones. The reaction of 1,7-enynyl amides, perfluoroalkyl iodides, alcohols and benzene-1,3,5-triyl triformate (TFBen) in PhCF_3_ and in the presence of PdCl_2_(Ph_3_P)_2_, DPEphos, NIS, and Cs_2_CO_3_ gave product **51** in high yields with excellent *E/Z* selectivity (Figure 52) [64]. In this reaction, TFBen was used as the CO source and alcohols when making the ester products. A reaction mechanism suggests that the *n*-C_4_F_9_ radical derived from *n*-C_4_F_9_I adds to the C=C double bond of amide followed by 6-*exo* cyclization, incorporation with the Pd-catalyst, CO insertion, and esterification with MeOH to afford product **51a**.

Benzene-linked 1,6-eneynyl ethers are a class of good substrates for radical difunctionalization. Li and coworkers reported a reaction of such substrates for the synthesis of dicarbonylated benzofurans in 2015. The reaction of benzene-linked 1,6-eneynyl ethers, 2,2,6,6-tetramethyl-1-piperidinyloxy (TEMPO), *t*-BuONO and O_2_ in DMF at 40 °C for 8 h gave product **52** in moderate-to-good yields (Figure 53) [65]. Two oxygen atoms were introduced to the product from O_2_ and TEMPO, respectively. *t*-BuONO is a key reagent which provides NO_2_ and NO after decomposition of HNO_2._ The reaction mechanism suggests that the addition of TEMPO to the C=C double bond of ethers followed by 5-*exo* cyclization, trapping of O_2_, oxidative cleavage of the N-O bond to release 2,6,6-tetramethyl-1-nitroso-piperidine, and O-O bond cleavage/isomerization to afford product **52a**.

An Ag-catalyzed reaction of 1,6-eneynyl ethers for the synthesis of sulfonyl-methylated benzofurans was reported by Wu, Jiang and their coworkers in 2017. The reaction of benzene-linked 1,6-eneynyl ethers and sodium sulfinates in CH_3_CN and in the presence of K_2_S_2_O_8_ and AgNO_3_ afforded product **53** in moderate-to-good yields (Figure 54) [66]. The reaction mechanism suggests that the sulfonyl radical generated from the oxidation of PhSO_2_Na adds to the C=C double bond of ethers followed by 5-*exo* cyclization, oxidation to cation, nucleophilic addition of H_2_O, and enol/ketone isomerization to give product **53a**.

In 2017, Kumar and coworkers reported a visible light-induced reaction for the synthesis of trifluoromethylacylated benzofurans, benzothiophenes, and indoles. The reaction of 1-ethynyl-2-(vinyloxy)-benzenes and CF_3_SO_2_Na in CH_3_CN/H_2_O using phenanthrene-9,10-dione (PQ) as a photoredox catalyst gave heterocycles **54** in good yields (Figure 55) [67]. The proposed reaction mechanism suggests that the CF_3_ radical, generated from CF_3_SO_2_Na with photo-activated PQ, adds to the C=C double bond of 1-ethynyl-2-(vinyloxy)-benzenes followed by 5-*exo* cyclization, electron transfer from PQH radical, H_2_O addition and deprotonation, resulting in product **54**.

A reaction of 1,6-eneynyl ethers for the synthesis of sulfonylacylated benzofurans was introduced by the Sun group in 2018. The reaction of oxygen-linked 1,6-enynes, DMSO and H_2_O in the presence of NH_4_I afforded product **55** in moderate-to-high yields (Figure 56) [68]. A reaction mechanism suggests that the reaction between DMSO and NH_4_I produced MeS and OH radicals. Addition of MeS radical to the C=C double bond of ethers followed by 5-*exo* cyclization, OH radical coupling, axidation of sulfide, and keto-enol tautomerism resulted in product **55a**.

In 2020, the Zhang group introduced a Pd-catalyzed radical oxidative aryldifluoroalkylation of benzene-tethered 1,6-enynes for the synthesis of difluoroalkylated benzofuran, benzothiophene, and indole derivatives. The reaction of 1,6-enynes, ethyl difluoroiodoacetate and arylboronic acids 1,4-dioxane or DCE under the catalysis of PdCl_2_(PhP_3_)_2_ and DPE-phos gave product **56** in moderate-to-good yields (Figure 57) [69]. The resultant products can be converted into aromatic five-membered rings **57** via Fe(OTf)_3_-catalyzed isomerization. A reaction mechanism suggests that the CF_2_CO_2_Et radical generated from ICF_2_CO_2_Et adds to the C=C double bond of 1,6-enyne followed by 5-*exo* cyclization to form **M-38** and then reacts with Pd^I^I to form intermediate **M-39**. Intermediate **M-39** could also be generated from **M-38** through iodine transfer with ICF_2_CO_2_Et and then with Pd^0^. Coupling **M-39** with phenylboronic acid finishes the reaction and gives product **56a**.

A Cu-catalyzed radical reaction of benzene-tethered 1,6-enynes for the synthesis of trifluoroethylated dihydrobenzofurans was reported by the Jiang group in 2019. The reaction of 1,6-enynes, Togni’s reagent, CO_2_ and amines in DMSO under the catalysis of CuSO_4_ gave products **58** in good yields (Figure 58) [70]. The proposed reaction mechanism suggests that the CF_3_ radical derived from the Togni’s reagent adds to 1,6-enynes followed by *5-exo* cyclization to form radical **M-41**. Then, it might have two pathways to form product **58a.** In path a, vinyl radical **M-41** is oxidized by Cu^II^ to a cation **M-42**, followed by trapping with carbamate anion to form **58a**. Alternatively, in path b, vinyl radical **M-41** reacts with CuSO_4_, CO_2_, and amine to form carbamato complex **M-43**, which leads to the formation of product **58a** after reductive elimination of the catalyst. 

Gao, Ying and their coworkers reported a Pd-induced radical reaction for the synthesis of difluoroalkyl- and alkenylphosphinyl-functionalized heterocycles in 2021. The reaction of 2-vinyloxy arylalkynes, ICF_2_CO_2_Et and diphenylphosphine oxides in DCE under the catalysis of PdCl_2_(PPh_3_)_2_ and Xantphos gave product **59** in good yields and stereoselectivity (Figure 59) [71]. A reaction mechanism suggests that the CF_2_CO_2_Et radical derived from ICF_2_CO_2_Et under the catalysis of Pd^II^ adds to the C=C double bond of 2-vinyloxy arylalkynes followed by 5-*exo* cyclization and iodine atom transfer from PdI, through the oxidative addition of Pd^0^ to vinyl iodide, formation of diphenylphosphine oxide complex, reductive elimination of Pd catalyst to give product **59a**.

Using benzene-tethered and carbonyl-containing 1,6-enynes as a substrate for Cu-catalyzed radical reaction for the construction of cyanotrifluoromethylated 1-indanones was introduced by the Jiang group in 2020. The reaction of benzene-tethered 1,6-enynes, Togni’s reagent and trimethylsilyl cyanide (TMSCN) under the catalysis of Cu(OTf)_2_ gave product **60** in good yields (Figure 60) [72]. A reaction mechanism suggests that the trifluoromethyl radical generated from Togni’s reagent under the catalysis of Cu^II^ adds to the C=C double bond of 1,6-enyne followed by 5-*exo* cyclization, formation of Cu^III^-complex containing CN, and reductive elimination of the Cu-catalyst to give product **60a**. By using benzene-tethered 1,7-enynes, the Jiang group extended the reaction scope for the synthesis of cyanotrifluoromethylated (*Z*)-3,4-dihydronaphthalen-1(2*H*)-ones **61** (Figure 61) [73].

A Cu-catalyzed radical for the synthesis of cyanoalkyl and ester-functionalized 1-indanones was introduced by the Jiang group in 2021. The reaction of 1,6-enynes, cyclobutanone oxime esters in DCE at 80 °C under the catalysis of CuBr and1,10-Phen gave product **62** in good yields (Figure 62) [74]. Both functional groups come from cyclic oxime esters. A reaction mechanism suggests that the γ-cyanoalkyl radical, generated from cyclic oxime ester via a SET process with Cu^I^L_n_, adds to the C=C double bond of 1,6-enyne followed by 5-*exo* cyclization, formation of a Cu^III^ complex containing the ester group, and reductive elimination Cu^I^L_n_ to give product **62a**.

A visible light-induced radical reaction of benzene-tethered 1,6-enynes for the synthesis of the thiosulfonylated pyrrolo[1,2-*a*]benzimidazoles was reported by the Chen group in 2021. The reaction of 1,6-enynes and PhSO_2_SPh in CH_3_CN under the photo catalysis of Na_2_-Eosin Y gave **63** in moderate-to-good yields (Figure 63) [75]. The reaction mechanism suggests that the sulfonyl radical derived from PhSO_2_SPh adds to the C=C double bond of 1,6-enynes followed by 5-*exo* cyclization and coupling with the SPh radical to afford product **63a**. 

The Tu and Jiang groups, in 2016, reported a radical reaction of 1,5-enynes for the synthesis of sulfonylated indeno[1,2-*d*]pyridazines. The reaction of 1,5-enynes, arylsulfonyl hydrazides in CH_3_CN and in the presence of I_2_ and TBHP gave products **64** in good yields (Figure 64) [76]. A reaction mechanism suggests that sulfonylhydrazone, generated from the condensation of 1,5-enynes with the arylsulfonyl hydrazide, reacts with the tosyl radical, which is also derived from arylsulfonyl hydrazide followed by 5-*exo* cyclization, 1,6-H atom transfer, 6-*endo* cyclization of the N-radical, and aromatization to give product **64a**.

A Pd-catalyzed radical cyclization of 1,7-enynes for the synthesis of functionalized (*E*)-3,4-dihydro-naphthalen-1(2*H*)-ones was reported by Jiang, Tu and their coworkers in 2018. The reaction of 1,7-enynes, sulfinic acids and *N*-fluorobenzenesulfonimide (NFSI) in THF under the catalysis of [Pd(CH_3_CN)_4_](BF_4_)_2_ gave **65** in good yields and high stereoselectivity (Figure 65) [77]. A possible reaction mechanism suggests that 1,7-enynes generate a Pd^II^ complex which then reacts with NFSI to form Pd^IV^ complex **M-44** for following two pathways. Under the reaction conditions for path a, complex **M-44** eliminates HBs_2_N, followed by the addition of R^3^SO_2_ radical, 6-*exo* cyclization, and reductive elimination of Pd catalyst to give fluorosulfonated product **65**. Under the reaction conditions for path b, HF is released from complex **M-44** followed by the similar reaction process of R^3^SO_2_ radical addition, 6-*exo* cyclization, and reductive elimination of Pd catalyst to give benzenesulfonylated products **66**.

Using of benzene-tethered 1,8-dienes for Ir-catalyzed oxidative difluorinative radical cyclization for the preparation of enol and CF_2_-containing benzoxepines was reported by the Yang group in 2018. The reaction of 1,8-dienes and BrCF_2_CO_2_Et in CH_2_Cl_2_/H_2_O under the photoredox catalysis with Ir(dtbbpy)(bpy)_2_PF_6_ afforded benzoxepine product **67** in good yields (Figure 66) [78]. A reaction mechanism suggests that the CF_2_CO_2_Et radical, generated from BrCF_2_CO_2_Et under the photocatalysis of Ir(dtbbpy)(bpy)_2_PF_6_, adds to the C=C double bond of 1,8-dienes followed by 7-*exo* cyclization, the formation of an iminium ion through the oxidization of [Ir^IV^(dtbbpy)(bpy)_2_PF_6_]^+^, and iminium hydrolysis to give product **67**.

Using unique benzene-tethered 1,5-enynes, the use of 4-(2-ethynylbenzylidene)cyclohexa-2,5-dien-1-ones for the synthesis of substituted spiroindene compounds was introduced by Yao in 2018. The reaction of 1,5-enynes, TMSN_3_ and NIS in dioxane in the presence of TBPB gave product **68** in good-to-excellent yields (Figure 67) [79]. The suggested reaction mechanism indicated that N_3_ radical derived from TMSN_3_ adds to the double bond of 1,5-enynes to give cyclohexadienone radical **M-45** (path a), which then undergoes 5-*exo* cyclization to form spirocyclic vinyl intermediates **M-46**, followed by iodine atom transfer from NIS to selectively give iodo- and azido-functionalized spiroindene products **68a** as an *E*-isomer. Due to the steric hindrance of **M-47**, cyclization through path b leading to the formation of *Z*-product **68a’** is unfavorable.

A metal-catalyzed radical spiroannulation of 1,5-enynes for the synthesis of fluorine-containing (*Z*)-spiroindenes was reported by Jiang’s group in 2020. The reaction of 1,5-enynes and ICF_2_CO_2_Et in DCE at 70 °C under the catalysis of PdCl_2_ and 9,9-dimethyl-4,5-bis(diphenylphosphino)xanthenes (Xant-Phos) gave iododifluoro-acetylated product **69** in good yields (Figure 68) [80]. However, the use of BrCF_2_CO_2_Et or C_4_F_9_I as the fluoroalkylation reagents failed to give the corresponding (*Z*)-spiroindenes. Another reaction of 1,5-enynes, Togni’s reagent and TMSCN in CH_3_CN at 50 °C under the catalysis of Cu(OAc)_2_ and 3,4,7,8-tetramethyl-1,10-phenanthroline (tmphen) gave trifluoromethylated products **70**. For the synthesis of **69a**, the reaction mechanism suggests that the CF_2_CO_2_Et radical derived from ICF_2_CO_2_Et adds to the C=C double bond of 1,5-enynes followed by 5-*exo* spirocyclization, formation of the Pd^II^-I complex, and reductive elimination of Pd catalyst to afford iododifluoroacetylated product **69a**. In the synthesis of CF_3_-functionalized products **70,** the CF_3_ radical derived from Togni’s reagent has a similar spirocyclization mechanism to form cyanotrifluoromethylated spiroindene product **70a**. The Tu and Jiang groups extended this reaction in the synthesis of iodosulfonylated spiroindenes, which involves an ionic instead of a radical cyclization [81].

Using dicyano-substituted benzene-tethered 1,5-enynes for a visible light-driven radical haloazidative cyclization for the synthesis of holoazido-functionalized indenes was accomplished by the Li group in 2020. The reaction of 1,5-enynes, TMSN_3_, and *N*-iodo (bromo or chloro) succinimide in DMF under the radiation of LED (380–385 nm) afforded product **71** in moderate-to-good yields (Figure 69) [82]. The suggested reaction mechanism indicated that the azide radical generated from TMSN_3_ under the photo conditions adds to the double bond of 1,5-enyne followed by 5-*exo* cyclization and I-atom transfer from NIS to give product **71a**.

Using benzene-tethered 1,7-diynes for the synthesis of iododifluoroacetal tetrahydronaphthalen-1-ols was introduced by the Jiang group in 2021. The reaction of 1,7-diynes and ICF_2_CO_2_Et under photoredox catalysis with *fac*-Ir(ppy)_3_ gave difluoromethyl-containing (1*E*,2*E*)-tetrahydronaphthalen-1-ols **72** bearing two exocyclic C=C double bonds as major stereoisomers in good yields (Figure 70) [83]. A reaction mechanism suggests that the CF_2_CO_2_E radical derived from ICF_2_CO_2_Et under the photocatalysis adds to the terminal alkyne of 1,7-diyne followed by 6-*exo* cyclization, SET of DIPEA to form cation, and nucleophilic addition with iodide anion to give (1*E*,2*E*)-product **72a** as a major isomer.

## 4. Reaction of Arene-Terminated Alkenes and Alkynes 

Presented in this section are the radical addition and cyclization-initiated difunctionalization reactions of arene-terminated alkenes and alkynes with a reaction sequence shown in Figure 71. For the class of substrates shown in Figure 72, the initial radical addition happens at the alkene or alkyne groups instead of the arene. Sequential radical cyclization leads to the formation of spiro- or fused-ring compounds. The only exception is the reaction of substrate **III-E**. The radical is added to the benzyne ring (via the benzyne intermediate). Among the general substrates, the reactions of alkynes **III-A** (arylpropiolamides if Y is NR) for making spiro compounds are much more popular than those of substrates **III-B** to **III-E** for making fused cyclic products. 

There are several reports on the reaction of arylpropiolamides for the synthesis of 3-functionalized azaspiro[4,5]trienones. In 2014, Li and co-workers reported a radical spirocyclization reaction of arylpropiolamides for the synthesis of 3-acylated azaspiro[4,5]trienones. The reaction of alkynyl amides and aldehydes in the presence of TBHP gave product **73** in good-to-excellent yields (Figure 73) [84]. The reaction mechanism suggests that the carbonyl radical generated from aldehyde adds to alkyne followed by *ipso*-carbocyclization, coupling with OH radical and oxidation of OH group to give 3-acylspiro[4,5]trienone **73a**. In 2014, Li’s group also reported a Cu-catalyzed radical spirocyclization of aryl alkynyl amides for the synthesis of azaspiro[4,5]trienones. The reaction of arylpropiolamides and cyclic ethers in *t*-BuOAc under the catalysis of Cu^II^ and TBHP gave product **74** in good yields (Figure 74) [85].

A Cu-catalyzed radical spirocyclization of arylpropiolamides for the synthesis of 3-triflouromrthylated azaspiro[4,5]trienones was reported by the Liang group in 2015. The reaction of alkynyl amides and NaSO_2_CF_3_ (Langlois’ reagent) in CH_3_CN in the presence of TBHP, MnO_2_ and CuCl gave product **75** in good-to-excellent yields (Figure 75) [86]. The reaction mechanism suggests that the CF_3_ radical derived from the Langlois’ reagent adds to the C≡C triple bond followed by *ipso*-carbocyclization, coupling with the *t*-BuOO radical, and elimination of *t*-BuOH to give product **75a**.

In 2015, the Wang group introduced an Ag-catalyzed radical spirocyclization of arylpropiolamides for the construction of 3-arylthiolated azaspiro[4,5]trienones. The reaction of alkynyl amides, thiophenols and H_2_O in 1,4-dioxane under the catalysis Ag^I^ gave product **76** in moderate-to-good yields (Figure 76) [87]. A proposed reaction mechanism suggests that the thiyl radical produced from thiophenol adds to the carbon triple bond of arylpropiolamides followed by the *ipso*-carboncyclization, SET to form carbocation, nucleophilic addition of H_2_O, and oxidization of OH to give product **76**. 

A TEMPO-mediated radical nitrative spirocyclization of arylpropiolamides for the preparation of 2-nitrated azaspiro[4,5]trienones was introduced by Li’s group in 2015. The reaction was carried out using arene-terminaled 1,5-enynes and *t*-BuONO in EtOAc in the presence of O_2_ and TEMPO to give nitrated spiro compound **77** in moderate-to-good yields (Figure 77) [88]. A reaction mechanism suggests that NO_2_ generated from the oxidization of NO adds to the carbon triple bond of arylpropiolamide followed by *ipso*-carbocyclization, TEMPO oxidation to form cation, nucleophilic addition of H_2_O, and oxidization to give product **77a**.

In 2015, Wang and co-workers developed an oxidative radical spirocyclization reaction of arylpropiolamides for the preparation of 3-sulfonated azaspiro[4,5]trienones. The reaction of arylpropiolamides and sulfonylhydrazide in the presence of TBHP and I_2_O_5_ afforded product **78** in moderate-to-good yields (Figure 78) [89]. The reaction mechanism suggests that the sulfonyl radical derived from sulfonylhydrazide adds to the carbon triple bond of amides followed by *ipso*-cyclization, SET to form cyclohexadienyl cation, nucleophilic addition of H_2_O, and finally oxidation with TBHP to give product **78**.

A new method for radical spirocyclization of arylpropiolamides to synthesize 3-sulfonated azaspiro[4,5]trienones was reported by Liu’s group in 2016. The reaction of amides and AgSCF_3_ in CH_3_CN in the presence of K_2_S_2_O_8_ and TBHP gave product **79** in excellent yields (Figure 79) [90]. A proposed reaction mechanism suggests that the CF_3_S radical derived from AgSCF_3_ adds to the carbon double bond of amides, followed by *ipso*-carbocyclization, coupling with *t*-butylperoxy radical, and elimination of *t*-BuOH to give product **79a**.

Other than the reactions of arylpropiolamides for making the spiro compounds described above, the reactions of *N*-phenylacrylamides have also been developed for making fused-cyclic products. In 2022, Zhang and co-workers reported a Co-promoted reaction for the synthesis of bromoarylthiolated heterocyclic compounds. The reaction of *N*-arylacrylamides and disulfides in CH_3_CN in the presence of CoBr_2_ and (NH_4_)_2_S_2_O_8_ gave functionalized product **80** in good-to-excellent yields (Figure 80) [91]. The reaction mechanism suggests that bromine and PhS radicals for the difunctionalization are generated from the reaction of CoBr_2_ and PhSSPh. The PhS radical adds to the terminal carbon of the double bond of amides, followed by cyclization and bromo radical coupling to give product **80a**.

The reaction of methacryloyl benzamides could result in six-membered ring-fused products. This work was reported by Tang, Chen and their co-workers in 2016 in the development of a Cu-catalyzed radical reaction for the synthesis of dicyanoisoproylated isoquinolinediones. The reaction of methacryloyl benzamides and AIBN in dioxane in the presence of CuI, KF, and K_3_PO_4_ gave product **81** in good-to-excellent yields (Figure 81) [92]. The reaction mechanism suggests that homolytic cleavage of AIBN gives two CNMe_2_C radicals. One of them adds to the carbon double bond of amides, followed by 6*-exo* cyclization to the benzene ring, selectively trapping the second CNMe_2_C radical under the assistance of CuI, and final step aromatization to give isoquinoline-1,3(2*H*,4*H*)-dione **81a**.

The reaction of *N*-propargylindoles could result in the formation of products with a core of 9*H*-pyrrolo[1,2-*a*]indol-9-one. In 2022, Du and coworkers developed photoredox radical cyclization of *N*-propargylindoles for the synthesis of 2-substituted 9*H*-pyrrolo-[1,2-*a*]indol-9-ones. The photo reaction of *N*-propargylindoles and cyclic ethers in MeCN at 80 °C in the presence TBHP and dual catalysts Cu(OAc)_2_ and Eosin Y give product **82** in moderate yields (Figure 82) [93]. The proposed mechanism suggests that a THF radical, generated from the reaction of THF with TBHP and the catalysts, adds to the carbon triple bonds of *N*-propargylindoles followed by 5-*exo* cyclization to give intermediate **M-48**. Intermediate **M-48** could have three paths to give product **82a**, (1) **M-48** couples with *t*-BuOO radical and then oxidation; (2) **M-48** traps O_2_ then reacts with TBHP and CuI catalyst; (3) **M-48** oxidized to cation through SET process and then oxidized OH to C=O.

Other than the addition of an initial radical to the alkene or alkyne group on the side chain presented in previous cases, a radical could add to benzene if the ring is converted to a benzyne. In 2021, the Studer group reported such a reaction in the synthesis of substituted five-membered heterocycles. The reaction of arenes bearing 1,2-TMS and OTs groups with TEMPO in the presence of CsF and 18-crown-6 ether gave product **83** in moderate yields (Figure 83) [94]. A proposed reaction mechanism suggests that arene is first converted to benzyne with the treatment of CsF and then reacts with TMPO radical followed by *5-exo* cyclization and coupling with the second TEMPO to give product **83a**. 

## 5. Reaction of Other Alkene and Alkyne Compounds

Presented in this section are the radical addition-initiated difunctionalizations of alkene- and alkyne-related compounds that cannot be fit in the previous sessions in terms of substrates or reaction mechanism. As shown in Figure 84, substrates **IV-A** to **IV-C** are 1,n-eneallenes; the cyano group in enenitrile **IV-D** is responsible for the second functionalization; arene-terminated enyne **IV-E** has a preexisting MeO group on the benzene ring which will be converted to a new functional group during the reaction; arene-terminated **IV-F** has a leaving group X which will be displaced by a new group at the step of second functionalization. Since the reactions of these substrates are not the major focus of this paper, only selected examples are highlighted.

An early example of radical difunctionalization of eneallenes was reported by the Hatem group in 1995 for the synthesis of bromo- and tosyl-functionalized cyclopantenes. The reaction of eneallenes and tosyl bromide in benzene using AIBN as a radical initiator gave product **84** (Figure 85) [95]. A proposed reaction mechanism suggests that the tosyl radical generated from TsBr adds the central carbon of allene, followed by 5-*exo* cyclization and coupling with bromine radical, to give product **84a**. Addition of tosyl radical to alkene instead of allene could be possible. However, since no expected product **84a’** was isolated, path b is less favorable than path a.

A later example for the reaction of eneallenes was reported by the Ma group in 2012. It is a Zn-catalyzed radical cyclization for the synthesis of iodoperfluoroalkylated five-membered rings. The reaction of eneallenes and R_F_I in CH_2_Cl_2_ in the presence of Zn powder and HOAc gave product **85** in moderate-to-good yields (Figure 86) [96]. It is worth mentioning that the two diastereomers of the product **85** could be converted into 3-(1-enylidene)heterocyclopentanes **86** through the TBAF-promoted dehydroiodination reaction. A mechanism for the racial reaction suggests that the perfluoroalkyl radical generated from R_F_I adds to the alkene carbon of eneallenes followed by 5-*exo* cyclization and coupling with the iodine radical from R_F_I to give product **85**.

A more recent example of eneallene reaction was reported by the Shi group in 2021. It is a visible light-induced radical reaction of ene-vinylidenecyclopropanes (ene-VDCP) for the synthesis of iodoperfluoro-alkylated *N*-heterocycles. The reaction of ene-VDCP, ICF_2_CO_2_Et or ICF_2_CF_2_CF_2_CF_3_ in 1,4-dioxane under the blue LED photocatalysis with *fac*-Ir(ppy)_3_ gave **87** in good yields and stereoselectivity (Figure 87) [97]. The reaction mechanism suggests that the CF_2_CO_2_Et radical, generated from ICF_2_CO_2_Et under the photolysis, adds to the terminal carbon of alkene followed by 5-*exo* cyclization, cyclopropane ring-opening, and extraction of iodine atom from ICF_2_CO_2_Et to give the final product **87a**.

An interesting example of using the cyano group as a radical acceptor for the difunctionalization reaction was reported by the Li group in 2015. It is a Cu-catalyzed radical cyclization of arene-tethered enenitrile for the synthesis of substituted quinoline-2,4(1*H*,3*H*)-diones. The reaction of *o*-cyanoarylacrylamide and diphenyl-phosphine oxide in CH_3_CN in the presence of CuBr_2_ and Mg(NO_3_)_2_·6H_2_O gave phosphinylated quinoline-2,4(1*H*,3*H*)-diones **88** in good-to-excellent yields (Figure 88) [98]. The reaction mechanism suggests that the Ph_2_P(O) radical derived from Ph_2_P(O)H under Cu^II^ catalysis adds to the C=C double bond of amide followed by 6-*exo* cyclization to the CN group and hydrolysis with H_2_O to give final product **88a**. 

In 2016, the Li group also reported a decarboxylative radical reaction of *o*-cyanoarylacrylamides for the preparation of carbonylated quinoline-2,4(1*H*,3*H*)-diones. The reaction of *o*-cyanoarylacrylamide and *α*-keto acids in acetone-H_2_O at 120 °C under the catalysis of AgNO_3_ and (NH_4_)_2_S_2_O_8_ gave product **89** in good yields (Figure 89) [99]. 

Having a MeO group on the benzene ring is a useful synthetic approach to assist radical cyclization and for dearomatization. In 2017, Li and co-workers developed a Ni-promoted radical spirocyclization of *N*-(*p*-methoxyaryl)propiolamides for the synthesis of 3-substituted azaspiro[4,5]trienones. The reaction of amides and *α*-bromo esters in DMF in the presence of Ni(acac)_2_, 1,2-bis(diphenylphosphino)ethane (dppe), TBHP and K_2_HPO_4_ gave product **90** in moderate yields (Figure 90) [100]. A proposed mechanism suggests that alkyl radical derived from *α*-bromo esters adds to the triple bond of amide followed by *ipso*-carbocyclization, oxidation with TBHP to form oxonium cation, and a final step of demethylation to give product **90a**. The MeO group on the aromatic ring is critical for the radical cyclization and formation of the carbonyl group through diaromatization. The product generated from this method is similar to that presented in Figure 73, in which there is no preexisting MeO group on the benzene ring.

Using a similar synthetic strategy and the alkyne substrate, in 2018, Liu and co-workers reported a visible light-mediated radical spirocyclization of *N*-(*p*-methoxyaryl)-propiolamides for the synthesis of 3-acylspiroc (Figure 91) [101]. The photo reaction of alkynes and benzoyl chloride in CH_3_CN in the presence of Ir^III^(ppy)_3_ and 2,6-lutidine gave product **91** in good-to-excellent yields.

Figure 92 shows another example of the reaction of *N*-(*p*-methoxyaryl)-propiolamides developed by Liu’s group also for the synthesis of 3-acylspiro[4,5]trienones [102].The photoredox reaction of alkynes, acyl oxime esters, H_2_O under the catalysis of Ir(ppy)_3_ gave product **92** in good yields.

The last example in this section is the reaction of arene-terminated alkene, which has a leaving group X on the aromatic ring. Liao and coworkers employed this substrate in the synthesis of functionalized benzosultams. The reaction of *N*-(2-haloaryl)cyanamide, bromodifluoroalkyl reagents and Na_2_S_2_O_5_ in DMF and H_2_O at 80 °C afforded product **93** in good yields (Figure 93) [103]. A proposed reaction mechanism suggests that the CF_2_CO_2_Et radical derived from BrCF_2_CO_2_Et SO_2_ adds to the carbon double bond of amide followed through 5-*exo* cyclization to the CN group, capture of SO_2_ (generated from Na_2_S_2_O_5_) to form sulfonyl radicals, cyclization to the benzene ring at the carbon with iodine, and a last step of deiodo aromatization to give product **93a**. 

## 6. Conclusions

Radical reactions are powerful and versatile synthetic methods for making carbon–carbon and carbon–heteroatom bonds. Designing one-pot and cascade radical transformations to make cyclic ring skeletons are highly efficient and operationally straightforward methods. Summarized in this article are the radical addition followed by cyclization reactions to make difunctionalized cyclic molecules. The second functionalization could be achieved through radical coupling, transition metal-assisted reaction, and nucleophilic or electrophilic substitution reactions, which significantly broaden the scope of difunctionalization reactions. Reactions of substrates such as dienes, diynes, and enynes, as well as of their arene-bridged and terminated analogs, are presented. In addition to conventional radical reactions using radical initiators or under transition metal-catalysis, the recent development of photoredox and electrochemical reactions have enhanced the scope of the radical difunctionalizations. In addition to the difunctionalization of unsaturated carbons such as alkenes and alkynes, we expect to see more development on difunctionalization reactions involving other functional groups, such as CN and N_3._ We also expect to see more applications in the synthesis of biologically significant molecules and natural products.

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
