# Peer review of "Difunctionalization of Dienes, Enynes and Related Compounds via Sequential Radical Addition and Cyclization Reactions"

_molecules, 2023, doi:10.3390/molecules28031145_

Round 1

Reviewer 1 Report

In this review, the authors have described radical reactions to make C-C and  C-heteroatom bonds. This reaction can be useful to difunctionalise dienes, diynes and enynes as well as their arene-bridged and terminated analogues.

In this review, different routes have been described, "in addition to conventional radicals", reactions using radical initiators or under transition metal catalysis, and the recent development of photoredox and electrochemical reactions.

The review is well written and the reactions are well described, for each reaction the mechanism is developed.

Before publishing, I suggest that the authors read the review carefully to correct errors in examples:

Page 2: line 52: delete the "e" are e more

Page 4: Renaud's paper was published in 2008 and not in 2014.

Page 6: Pd(PPh3)2Cl2 not Pd(PPh3) correct also in figure 10.

Page 7: the authors of reference 24 said "radical cyclization of 1,6-enynes for the synthesis of substituted pyrrole derivatives" in my opinion these are pyrrolidine derivatives not pyrrole.

Page 21: scheme 36 "Bpy" not Ru(bby).

Page 33: add v to the visible scheme 55.

 I also have some comments on the electrochemistry for radical cyclisation, in this review the authors have described only one example. Please add more examples:

like this article: "Electrochemical annulation-iodosulfonylation of para-quinone methides (p-QMs) containing 1,5-Enyne to access (E)-Spiroindenes" Org. Lett. 2020, 22, 11, 4471-4477, DOI: 10.1021/acs.orglett.0c01470

You must cite these reviews: "Multi-component reactions and photo/electrochemistry join forces: atom economy".
electrochemistry join forces: atom economy meets energy efficiency" Chem. Soc. Rev. 2022, 51, 2313-2382

"Radical cyclization of 1,n-Enynes and 1,n-Dienes for the synthesis of 2-Pyrrolidone" ChemAsianJ.2021,16, 3068-308.

Author Response

In this review, the authors have described radical reactions to make C-C and  C-heteroatom bonds. This reaction can be useful to difunctionalise dienes, diynes and enynes as well as their arene-bridged and terminated analogues.

In this review, different routes have been described, "in addition to conventional radicals", reactions using radical initiators or under transition metal catalysis, and the recent development of photoredox and electrochemical reactions.

The review is well written and the reactions are well described, for each reaction the mechanism is developed.

Before publishing, I suggest that the authors read the review carefully to correct errors in examples:

Page 2: line 52: delete the "e" are e more

Page 4: Renaud's paper was published in 2008 and not in 2014.

Page 6: Pd(PPh3)2Cl2 not Pd(PPh3) correct also in figure 10.

Page 7: the authors of reference 24 said "radical cyclization of 1,6-enynes for the synthesis of substituted pyrrole derivatives" in my opinion these are pyrrolidine derivatives not pyrrole.

Page 21: scheme 36 "Bpy" not Ru(bby).

Page 33: add v to the visible scheme 55.

Answer: Thanks for referee’s positive comments. The typos are fixed.

I also have some comments on the electrochemistry for radical cyclisation, in this review the authors have described only one example. Please add more examples:

like this article: "Electrochemical annulation-iodosulfonylation of para-quinone methides (p-QMs) containing 1,5-Enyne to access (E)-Spiroindenes" Org. Lett. 2020, 22, 11, 4471-4477, DOI: 10.1021/acs.orglett.0c01470

Answer: This paper is cited as ref 81.   Following sentence added to the MS (p30). The Tu and Jiang extended this reaction in the synthesis of iodosulfonylated spiroindenes which involves an ionic instead of a radical cyclization [81].

You must cite these reviews: "Multi-component reactions and photo/electrochemistry join forces: atom economy".
electrochemistry join forces: atom economy meets energy efficiency" Chem. Soc. Rev. 2022, 51, 2313-2382

Answer: This paper is cited as ref 16.

"Radical cyclization of 1,n-Enynes and 1,n-Dienes for the synthesis of 2-Pyrrolidone" ChemAsianJ.2021,16, 3068-308.

Answer: This paper is cited as ref 45.

Reviewer 2 Report

The authors present in this work the difunctionalization of dienes, enynes and other similar compounds via sequential radical addition and cyclization reactions. The compilation of different methodologies and reactions has been a wonderful work, perfectly appropriate for the knowledge of the reader. I would like to highlight that the number of references is suitable and in agreement with the works described in the article.

Despite the fact that the high number of pages, the read is easy for the reader and I consider that the work satisfies the main objective of this review, hence, I consider that the article is perfectly suitable for publication in Molecules.

I have only one suggestion, some methodologies described in the work are under conventional conditions with high reaction times. If it is possible, it should be interesting compare this methodologies with synthesis under microwave irradiation because of its advantages. Microwave irradiation improves the yields, and decrease the reaction times. If there are some examples under microwave irradiation with the same purposes, please include them. If not, it is ok in this way.

Author Response

The authors present in this work the difunctionalization of dienes, enynes and other similar compounds via sequential radical addition and cyclization reactions. The compilation of different methodologies and reactions has been a wonderful work, perfectly appropriate for the knowledge of the reader. I would like to highlight that the number of references is suitable and in agreement with the works described in the article.

Despite the fact that the high number of pages, the read is easy for the reader and I consider that the work satisfies the main objective of this review, hence, I consider that the article is perfectly suitable for publication in Molecules.

I have only one suggestion, some methodologies described in the work are under conventional conditions with high reaction times. If it is possible, it should be interesting compare this methodologies with synthesis under microwave irradiation because of its advantages. Microwave irradiation improves the yields, and decrease the reaction times. If there are some examples under microwave irradiation with the same purposes, please include them. If not, it is ok in this way.

Answer: Thanks for author’s positive comments. The radical reactions are commonly kinetically controlled which may have several competitive pathways. Microwave reactions could create “superhot” environment in the reaction system which may not be favorable for the desired pathway (Angew Chem Int Ed 2013,1088, 10.1002/anie.201204103).  It could be a reason why there are not many microwave-heated radical reactions in literature.

Reviewer 3 Report

This review article summarizes the recent development of reactions involving radical addition and cyclization of dienes, diynes, and enynes, as well as arene-bridged and arene-terminated compounds for the preparation of difunctionalization cyclic compounds. The present work provides plenty of examples with detailed and comprehensive references. The discussions are nicely organized. I believe it is a very important topic that needs to be covered, and the manuscript provides an excellent account of the related work in this area. Therefore, I recommend acceptance of this impressive review paper in Molecules after minor revisions as described below:

1, Please unify the size of the schemes, for example, scheme 1 and scheme 2; scheme 3 and scheme 4.

2, Please include the stereochemistry (ZE ratio) for 1c, 1d, 1e in scheme 5 according to the ref. 19 in the original TL paper the authors cited.

Author Response

This review article summarizes the recent development of reactions involving radical addition and cyclization of dienes, diynes, and enynes, as well as arene-bridged and arene-terminated compounds for the preparation of difunctionalization cyclic compounds. The present work provides plenty of examples with detailed and comprehensive references. The discussions are nicely organized. I believe it is a very important topic that needs to be covered, and the manuscript provides an excellent account of the related work in this area. Therefore, I recommend acceptance of this impressive review paper in Molecules after minor revisions as described below:

1, Please unify the size of the schemes, for example, scheme 1 and scheme 2; scheme 3 and scheme 4.

2, Please include the stereochemistry (ZE ratio) for 1c, 1d, 1e in scheme 5 according to the ref. 19 in the original TL paper the authors cited.

Answer: Thanks for author’s positive comments. The size for Schemes 1-4 and ZE ratio in Scheme 5 are fixed.